# Spectral Form Factor and Dynamical Localization

**DOI:** 10.3390/e25030451

**Published:** 2023-03-04

**Authors:** Črt Lozej

**Affiliations:** Max Planck Institute for the Physics of Complex Systems, Nöthnitzer Str. 38, 01187 Dresden, Germany; crt@pks.mpg.de

**Keywords:** quantum chaos, spectral form factor, dynamical localization, billiards, stadium

## Abstract

Quantum dynamical localization occurs when quantum interference stops the diffusion of wave packets in momentum space. The expectation is that dynamical localization will occur when the typical transport time of the momentum diffusion is greater than the Heisenberg time. The transport time is typically computed from the corresponding classical dynamics. In this paper, we present an alternative approach based purely on the study of spectral fluctuations of the quantum system. The information about the transport times is encoded in the spectral form factor, which is the Fourier transform of the two-point spectral autocorrelation function. We compute large samples of the energy spectra (of the order of $10^{6}$ levels) and spectral form factors of 22 stadium billiards with parameter values across the transition between the localized and extended eigenstate regimes. The transport time is obtained from the point when the spectral form factor transitions from the non-universal to the universal regime predicted by random matrix theory. We study the dependence of the transport time on the parameter value and show the level repulsion exponents, which are known to be a good measure of dynamical localization, depend linearly on the transport times obtained in this way.

## 1. Introduction

One of the central areas of study in quantum chaos is that of the spectral statistics of quantum chaotic systems and how they relate to classical chaos and random matrix theory (RMT) [1,2]. The spectral form factor (SFF) is one of the most widely used spectral statistics due to the stark contrast in behaviour between the chaotic and integrable regimes. However, the SFF is not a self-averaging quantity [3], meaning that the typical value may be far from the average value. Because of this, its numerical computation remains challenging, and its practical evaluation requires some sort of smoothing procedure, either by computing disorder averages (only possible when considering systems with disorder) or local time averages. Nevertheless, the SFF has been used as the fundamental indicator of quantum chaos in many of the central rigorous results. A heuristic proof of the quantum chaos (Bohigas–Giannoni–Schmit) conjecture [4,5], which was initiated by Berry [6], developed by Sieber and Richter [7], and later completed by Haake’s group [8,9,10], clearly relates random matrix spectral correlations to correlations among classical unstable (hyperbolic) orbits by computing the orbit contributions to the SFF. Recently, much attention has been given to the SFF in many-body settings [11,12,13,14,15,16]. Rigorous proofs of quantum chaos by computing the SFF have been preformed in kicked spin chains [17,18] and more generally in dual-unitary circuits [19,20]. In the high-energy physics context, for example, studies of the SFF have been performed in Sachdev–Ye–Kitaev-type models [21,22,23,24] and using hydrodynamic theories [25]. Pioneering experimental studies of the SFF were carried out on excitation spectra of molecules [26] and microwave billiards [27]. The SFF has recently been used to probe the many-body localization (MBL) transition [28,29]. In this paper, we will adapt a similar methodology to that in Ref. [28] to study the dynamical localization transition in single-body systems on the example of stadium billiards. Even though quantum billiards are ubiquitous in the field of quantum chaos, not many theoretical studies of the SFF in chaotic billiards are to be found in the literature. Previous studies focus mainly on the (pseudo)integrable and closely related regime, such as rectangular billiards [30], including perturbations [31], barrier billiards [32,33,34] and Veech triangular [35] billiards. Recently, the SFF has also been computed in the case of generic triangular billiards [36], where it was demonstrated that the spectral statistics follow RMT, thereby extending the quantum chaos conjecture to strongly mixing systems without classical Lyapunov chaos.

The origin of the study of dynamical localization in the stadium billiards can be traced to the pioneering work of Borgonovi, Casati and Li [37], later continued by Casati and Prosen [38,39]. Quantum dynamical localization (DL) occurs when quantum interference stops the diffusion of wave packets. The phenomenon is analogous to the famous Anderson localization, but occurring in momentum space instead of the configuration space. The two can be explicitly related in the example of the quantum kicked rotor system [40]. The following heuristic argument explains when dynamical localization may be expected. The transition is governed by the ratio of two typical time scales, namely the transport time $t_{T}$, controlling the typical rate of diffusion, and the Heisenberg time $t_{H}$, which is the inverse of the mean level spacing. The discreetness of the quantum spectrum may only be resolved on time scales greater than the Heisenberg time. If $t_{T}>t_{H}$, we expect the interference will localize the wave packets in only part of the momentum space. On the other hand, if $t_{T}<t_{H}$, we expect the wave packet to encompass the full extent of the momentum space before any interference effects might stop the diffusion. The transition from the dynamically localized regime to the fully delocalized ergodic regime has been extensively studied in the quantum kicked rotor system (see the review articles [41,42]), billiard systems [37,38,39,43,44,45,46,47,48], Dicke model [49], etc. In particular, our previous studies of DL in the stadium billiard [45] show the functional dependence of the localization measures and level repulsion exponents on the ratio $\alpha =t_{H}/t_{T}$. However, to ascertain the transport times, a separate classical computation of the transport times was necessary. This also introduces some ambiguity in defining the transport time because of the complex inhomogeneous diffusion that occurs (see Ref. [50] for details). Furthermore, in generic billiards with divided regular/chaotic phase space the diffusion process is even more complex because of the hierarchical structures of islands of stability in the phase space and the stickiness phenomenon (see Refs. [48,51,52,53] and references therein). In the present paper, we present an alternative definition of the transport time based on the timescale of the onset of RMT spectral statistics in the SFF. The definition is inspired by the methodology used to extract the Thouless time of spin chains in Ref. [28]. We show the transport time extracted from the spectral form factor can be used to describe the transition from the DL regime to the ergodic regime.

## 2. Definitions and Methods

### 2.1. Quantum Billiards

Quantum billiards are archetypical models of both classical and quantum chaos. In the quantum billiard problem, we consider a quantum particle trapped inside a region $B\subset R^{2}$ referred to as the billiard table. The eigenfunctions $\psi_{n}$ are given by the solutions of the Helmholtz equation
(1)$$\left(\nabla^{2}+k_{n}^{2}\right)\psi_{n}=0,$$
and Dirichlet b.c. $\psi_{n}{|}_{\partial B}=0$, with eigenenergies $E_{n}=k_{n}^{2}$, where $k_{n}$ is the wave number of the *n*-th eigenstate. We use a system of units where ℏ=1, and the mass of the particle is m=1/2. The very efficient scaling method, devised by Vergini and Saraceno [54] and extensively studied by Barnett [55], allows us to compute very large spectra of the order of $10^{6}$ states (the implementation is available as part of [56]). The spectral staircase function counts the number of eigenstates (or modes) up to some energy $N\left(E\right):=\#\left\{n|E_{n}<E\right\}$. The asymptotic mean of the spectral staircase for billiards is given by the well known generalized Weyl’s law [57]
(2)$$N_{\mathrm{Weyl}}\left(E\right)=\left(AE-L\sqrt{E}\right)/4\pi +c$$
where A is the area of the billiard, L is the circumference, and *c* is a constant corner and curvature correction. The asymptotic density of states is then
(3)$$\rho \left(E\right)=\frac{A}{4\pi }-\frac{L}{8\pi \sqrt{E}}.$$ The Heisenberg time is defined as the inverse of the mean level spacing or
(4)$$t_{H}=2\pi \rho \left(E\right).$$ To compare the universal statistical fluctuations it is convenient to unfold the spectra. This is performed by inserting the numerically computed billiard spectrum into Weyl’s formula $e_{n}:=N_{\mathrm{Weyl}}\left(E_{n}\right)$. The resulting unfolded spectrum $e_{n}$ has a uniform mean level density equal to one. In the unfolded spectrum $t_{H}=2\pi$.

One of the paradigmatic examples is the stadium billiard of Bunimovich [58,59]. The stadium is constructed from two semicircles separated by a rectangular region. We fix the radius of the semicircles to one. The family of stadium billiards is characterized by the width of the separation ε. The stadium is classically chaotic for any value of ε. Because of the two reflection symmetries, it is sufficient to consider the quarter stadium in the quantum case, corresponding to the odd–odd symmetry sector of the full stadium. Two examples of stadium eigenstates are shown in Figure 1. In panel (a), we show a typical dynamically localized eigenstate in the ε=0.02 stadium. The localization is evident in the distinctly regular nodal patterns that are similar in appearance to very strong scarring. Although the probability density function extends over all the configuration space, it is visibly depleted in the inner part of the billiard near the origin (note the colour scale is logarithmic). In (b), we show a typical eigenstate in the ε=0.5 stadium. The state is practically uniformly extended, with the typical chaotic nodal patterns of random superpositions of plane waves, with some scarring visible around an unstable (bow-tie-shaped) periodic orbit.

### 2.2. Spectral Form Factor

The SFF is loosely defined as the Fourier transform of the spectral two-point correlation function and may be written as
(5)$$K\left(\tau \right)=\langle {\left|\sum_{n}\exp\left(2\pi ie_{n}\tau \right)\right|}^{2}\rangle ,$$
where the sum goes over the unfolded energy levels. The time τ is measured in units of Heisenberg time $\tau_{H}=1$. The SFF is not a self-averaging quantity [3], it exhibits erratic fluctuations with time. This means a separate averaging must be performed, represented by 〈⋯〉. This is commonly an average over different realizations when considering random matrices or disordered systems. For clean single-body systems, we instead perform a moving time average to smooth out the fluctuations [26,27]. This is achieved by convolving the SFF with a Gaussian function in time,
(6)$$K\left(\tau \right)=\int_{0}^{\infty }{\left|\sum_{n}\exp\left(2\pi ie_{n}\tau \right)\right|}^{2}\frac{1}{\sqrt{2\pi \sigma^{2}}}\exp\left(-\frac{1}{2}\frac{{\left(\tau -t\right)}^{2}}{\sigma^{2}}\right)dt.$$ This introduces an additional numerical parameter σ. It is further useful to decompose the SFF into the connected and disconnected part $K=K_{\mathrm{conn}}+K_{\mathrm{disc}}$. The disconnected part is given by the diagonal terms from definition (Equation 5) and depends solely on the density of states (see Ref. [25] for more details). It is also evident from definition (Equation 5) that the SFF behaves as a delta distribution at t=0. This narrow peak is produced by the disconnected part of the SFF. The spectral fluctuations are encoded in the connected part of the SFF, which we obtain by subtracting the disconnected part $K_{\mathrm{conn}}=K-K_{\mathrm{disc}}$. Since we are only interested in spectral fluctuations, we will only consider the connected part of the SFF in all further instances.

The stadium billiards are classically chaotic systems with time-inversion symmetry. Their universal spectral statistics are therefore expected to follow the Gaussian orthogonal ensemble (GOE) of RMT [1,2]. In the infinite dimensional GOE case, the SFF has the following analytical form,
(7)$$K_{\mathrm{GOE}}\left(\tau \right)=\left\{\begin{array}{cc} 2\tau -\tau \ln(2\tau +1) & \tau <1\\ 2-\tau \ln(\frac{2\tau +1}{2\tau -1}) & \tau >1 \end{array}\right..$$ This has the basic anatomy of a so-called “ramp” followed by a saturation regime after reaching the Heisenberg time. This contrasts well with the integrable case, where an immediate saturation is expected. Since all stadium billiards are ergodic chaotic systems, we expect the SFF will follow the universal GOE prediction. However, when ε is small, the transport times become very large and should even diverge as we approach the limit ε→0 (the limiting case is the integrable circle billiard, where the momentum becomes a strictly conserved quantity). Classically, the fact that the system is ergodic becomes apparent only after the transport time is reached, and the dynamics are able to explore all the phase space. We expect the SFF of the stadia will follow the GOE prediction only after the transport elapses. We will therefore define the quantum transport time $\tau_{T}$ as the time at which the SFF of the numerically computed billiard spectrum begins to follow the RMT prediction. The procedure that is used to extract $\tau_{T}$ is described in more detail in Appendix A. The transport time may either be greater or smaller than $\tau_{H}=1$ (note that by definition (Equation 5) we measure time in the SFF in units of Heisenberg time). Following the argument from the introduction, this means we expect localization when $\tau_{T}>1$, and no localization (extendedness) when $\tau_{T}<1$.

### 2.3. Dynamical Localization and Level Repulsion

We will measure the localization of the eigenstates indirectly by computing the level of the repulsion exponent of the spectra. The level repulsion exponent is defined by using the nearest-neighbour level spacing. An intuitive understanding of the connection between level repulsion and localization may be gained from the following heuristic picture. The eigenstates of chaotic systems in the non-localized regime are extended in the phase space, and there is a great deal of overlap between them. In the extreme case of full extendedness, the differences in the overlaps will stem purely from local fluctuations of the wave (or Husimi) functions. This means strong couplings between the consecutive states are possible and indeed very probable, leading to a gap in the eigenenergies. For an extremely simplified case, one may consider a two-level system, where the gap of the eigenenergies (avoided crossing) may be computed directly. On the other hand, dynamically localized states only occupy a smaller area of the phase space. If the consecutive states occupy different areas of the phase space, there will be essentially no overlap between them, with the couplings exponentially suppressed. This is more likely to happen if the eigenstates are more severely localized, leading to a much weaker level repulsion. The connection between localization and level repulsion has a strong foundation in our previous works and also related studies in different systems. In particular, in Ref. [45] we showed that the level repulsion exponents in the stadium billiards are proportional to the mean values of localization measures based on the Husimi representation of the eigenstates (for a recent study of the localization measures in more general divided phase space systems, see also [60]).

The level spacing is defined as the difference in energy between two consecutive levels in the unfolded spectrum $s_{i}=e_{i+1}-e_{i}$. The unfolding procedure guarantees that the mean level spacing is in unity. We also studied the probability density distribution P(s). The level repulsion is given by the behaviour of P(s) at small *s*, namely $P\left(s\right)\propto s^{\beta }$, where β is called the level repulsion exponent. Following the quantum chaos conjecture, the level spacing distribution of chaotic quantum systems is well described by the Wigner surmise obtained from RMT. In the GOE case, β=1 indicates linear level repulsion. On the other hand, integrable systems are expected to show Poissonian level statistics (Berry–Tabor conjecture) and no level repulsion β=0. In the localized regime, the distribution is not known analytically. Empirically, the level repulsion exponent changes from 0 to 1 as we transition from the severely localized to the delocalized chaotic regime. One of the most popular ways of describing the level spacing distribution in the transition region is to use the Brody distribution [61], which interpolates the two regimes
(8)$$P_{B}\left(S\right)=cS^{\beta }\exp\left(-dS^{\beta +1}\right),$$
where the normalization constants are given by c=(β+1)d, and $d={\left(\Gamma \left(\frac{\beta +2}{\beta +1}\right)\right)}^{\beta +1}$. Alternatively, another popular choice is the Izrailev distribution [41]; however, we opted for the Brody distribution due to the simpler expression and empirically good description of the numerical results in previous papers [43,44,45,46,62]. The level repulsion exponent β is the indicator of dynamical localization, which we compare to $\tau_{T}$ across the transition.

## 3. Results

### 3.1. Transport Times

To compute the quantum transport times, we computed the spectra of the stadium billiards at 22 values of ε∈(0.01,0.07). Each spectrum contains around $10^{6}$ levels with $k_{n}\in \left(640,4000\right)$. The lowest levels start at around the $10^{4}$-th eigenstate. Because the scaling method computes the eigenvalues in only small intervals, some levels are lost in the computation due to numerical errors. By comparing with Weyl’s law, we estimate that less than 0.1% are lost.Since the SFF is a linear spectral statistic, we expect this to have a negligible effect on the result. Even with the great efficiency of the numerical method, collecting the spectra and computing the SFF takes considerable computational effort due to the large spectra required to obtain good results.

The connected SFF of the selected stadia are shown in Figure 2a. The numerical results are compared with the GOE curve (Equation 7). We see the ε=0.5 result, where the transport time is expected to be very short, nicely follows the GOE curve from start to finish. When ε is decreased, the numerical SFF detaches from the GOE curve at some point. This point is by our definition the transport time. We see the transport time increases as ε is decreased, eventually becoming longer than the Heisenberg time. We note the SFF still exhibits some fluctuations, even though each of the spectra contains many levels—approximately $10^{6}$. The smoothing parameter in the presented case is σ=0.01, which we find is the optimal compromise between fine resolution and the intensity of fluctuations. We extract the transport times, including some error estimates (shown with the error bars), as described in Appendix A. The result is presented in Figure 2b. In the inset, we show the same graph in the decadic log–log scale. The transport times appear to roughly follow a power law decay $\tau_{T}\propto \varepsilon^{-\gamma }$, with a transition from γ=1 to γ=1/2 above $\varepsilon_{c}\approx 0.04$. The caveat is that the power laws should not be seen as a definitive result, since the range of the parameter values is within one decade. In Ref. [50], we computed the classical transport times of the stadia in the space of conjugated momenta and discrete time (the conjugated momenta of the billiard mapping, describing the classical dynamics, are p=sinθ, where θ is the angle of reflection when the particle hits the boundary). There, we found $N_{T}\propto \varepsilon^{-\gamma }$ with a transition from γ=5/2 to γ=2 above $\varepsilon_{c}\approx 0.05$; however, the transitional value is not sharply defined. We note that considering the transport in the flow of the stadium billiard (real time) instead of the billiard map (discrete time) might give different results, because the slow decay of correlations in the classical stadium billiard is caused by special types of bouncing ball and boundary glancing orbits [63]. The difference in the decay rates indicates the quantum transport time extracted from the SFF is not directly proportional to the discrete transport time in momentum space. However, both are monotonic functions (within some fluctuations) of the parameter ε, and both seem to exhibit a transition in the power law behaviour at roughly the same parameter range.

### 3.2. Level Repulsion

To determine the level repulsion exponents β, we fit the level spacing distributions of the computed spectra with the Brody distribution (Equation 8). In Figure 3a, we show some examples of the fits. We see the level spacings are indeed described well by the Brody distribution. In panel (b), we show β as a function of $\tau_{T}$. We observe the transition from the extended to the localized regime as the transport time increases, empirically confirming the heuristic argument that the transition should happen when the transport time is close to the Heisenberg time. Quantitatively, the mid-point of the transition β=0.5 occurs already at $\tau_{T}\approx 0.8$. The relation between the two quantities appears to be close to linear. In Ref. [45], we found a nonlinear functional relation between β and the parameter $\alpha =t_{H}/t_{T}$ (the denominator is the classical transport time) that would be analogous to $1/\tau_{T}$. This indicates that the quantum transport times are not exactly analogous to the classical transport times. Nevertheless, we clearly establish a functional relation between the level repulsion exponent and the quantum transport times. Because the level repulsion exponents are a linear function of localization measures (see Refs. [45,46]), this demonstrates the link to dynamical localization and potentially also a more general relation (a similar non-universal but characteristic behaviour) between level spacing distributions and spectral form factors in other contexts.

## 4. Discussion

We have presented a numerical study of the spectral form factors of the stadium billiards in relation to dynamical localization. The main result is the computation of the connected spectral form factors and extraction of the quantum transport times $\tau_{T}$ (in units of Heisenberg time) from the SFF. By relating $\tau_{T}$ to the level repulsion exponent β, we show that the transition from the localized to the delocalized regime is governed by the ratio between the transport time and the Heisenberg time. The novelty of the presented approach compared with previous studies of the dynamical localization transition is that all computations are based on the quantum spectral statistics alone. No classical computations of the transport times are needed. This might be especially beneficial in cases where the classical transport processes are very complex and the definition of the relevant transport time might be ambiguous, such as, for instance, in systems with divided phase space and, as already demonstrated in Ref. [28], in many-body systems without a classical limit. The relationship between β and $\tau_{T}$ is close to linear. This is different from the nonlinear relation with the analogous quantity $\alpha =t_{H}/t_{T}$ found in Ref. [45], where the transport times $t_{T}$ were computed from the classical momentum diffusion in discrete time. Nevertheless, both definitions of the transport time exhibit a power law regime change at roughly the same value of ε. Since quantum billiards may be considered a generic example of Hamiltonian systems, the results are widely applicable. Further research directions might include a similar study of the SFF in systems with divided phase space, such as, for instance, the limaçon billiards (see Refs. [46,64,65] and references therein).

## Figures and Tables

**Figure 1 entropy-25-00451-f001:**
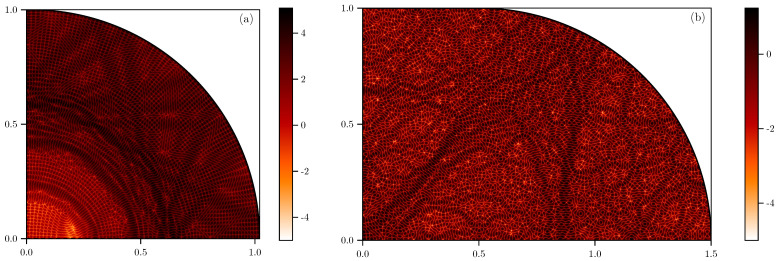
Representativeeigenstates in the (quarter) stadium billiards. (**a**) Localized eigenstate at k=302.60195 and ε=0.02. (**b**) Extended state at k=302.6037 and ε=0.5. We plot the probability density in the logarithmic scale, $\log_{10}{\left(|\psi \right|}^{2})$.

**Figure 2 entropy-25-00451-f002:**
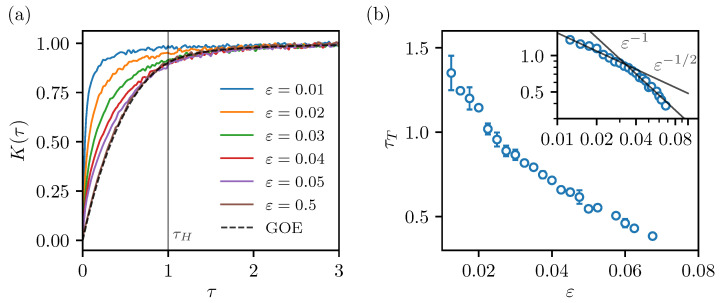
(**a**) Connected spectral form factors of stadium billiards in units of Heisenberg time. The GOE curve, expected in chaotic systems, is shown with the black dashed line. (**b**) Dependence of the quantum transport times (in units of Heisenberg time), extracted from the SFF, on the billiard parameter ε. The error bars show the estimated errors due to the fluctuations in crossing the threshold value. The inset shows the same plot in the decadic log–log scale.

**Figure 3 entropy-25-00451-f003:**
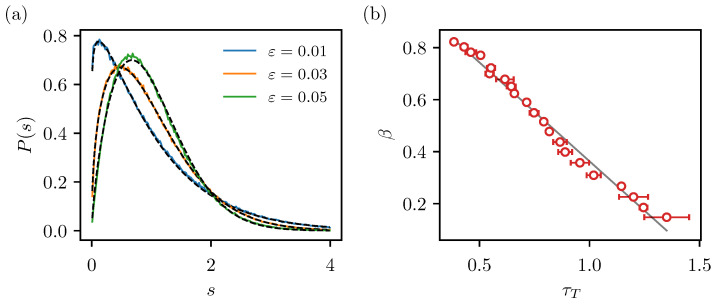
(**a**) Representative examples of nearest-neighbour level spacing distributions (coloured lines) fitted by the Brody distribution (black dashed lines). (**b**) Dependence of the level repulsion exponent (Brody parameter) on the quantum transport times (in units of Heisenberg time).

## Data Availability

The data presented in this study are available upon reasonable request from the corresponding author.

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
