# Peer review of "Spectral Form Factor and Dynamical Localization"

_entropy, 2023, doi:10.3390/e25030451_

Round 1

Reviewer 1 Report

The author studies the spectral form factor (SFF) in a family of stadium billiards and relates the transport time, i.e., the time after which universal random matrix behavior sets in, to the strength of nearest neighbor level repulsion. The later is related, by previous work of the author, to dynamical localization, i.e., localization in phase space of the billiard's eigenmodes. This provides the possibility to extract properties of the localization transition of eigenmodes directly from spectral statistics (the SFF) of the billiard.

The paper is well written, the results are presented clearly and are well supported by numerical evidence. I recommend the paper for publication after addressing a view minor remarks stated below.

1) The author introduces the concept of dynamcial localization and provides examples in terms of eigenmodes in real space. However, dynamical localization is detected by the strength of level repulsion obtained from distribution of level spacings only. Even though the author provides references, which demonstrate the connection between  localization and level repulsion I would like to see a bit more detail on this connection in the present manuscript. In particular, I was wondering whether there is an intuitive explanation of the connection between
localization and level repulsion.

2) As the spectral form factor around Heisenberg time probes spectral  correlations on energy scales corresponding to the mean level spacing, I was wondering, whether information about level repulsion (and hence localization) can be obtained directly from the SFF on this time scales?

3) The authors mention, that the transport times obtained from the SFF differ from those obtained by using the billiard dynamics in boundary phase space and that using real-time dynamics of the billiard flow might give results. Can transport times computed from real-time dynamics be related to those obtained by the discrete billiard map in a simple way, e.g., by using the mean time between boundary collisions? Would this bring the transport time obtained from closer to those extracted from SFF? I would expect the time-like variable on which the SFF depends to correspond to real time in the billiard dynamics, instead of discrete time.
(This remark stems from personal interested, as it concerns something  which is not directly in the scope of the manuscript.)

4) As there is some ambiguity in the definition of the transport time from the SFF by choosing g_0 I would like to know, if the scaling of transport times with deformation parameter presented in Fig. 2(b) depends on that choice?

Minor typos:
line 20... there is an "at" too much at the end of the line
line 40... the "of the" at the end of the line seems wrong

Notion of Heisenberg time seems not always consistent, e.g. t_H=1 in line 115 conflicts with Eq. (4) and line
96. I think some of the t_H should be called tau_H instead?

Author Response

I thank the reviewer for the careful reading of the manuscript and useful remarks which I address in the same order as they were presented:

1) I have added a small section of text presenting a heuristic picture of the connection between localization and level repulsion to subsection 3.2. I quote it here for convenience:

An intuitive understanding of the connection between level repulsion and localization may be gained from the following heuristic picture. The eigenstates of chaotic systems in the non-localized regime are extended in the phase space, and there is a great deal of overlap between them. In the extreme case of full extendedness, the differences in the overlaps will stem purely from local fluctuations of the wave (or Husimi) functions. This means strong couplings between the consecutive states are possible and indeed very probable, leading to a gap in the eigenenergies. For an extremely simplified case, one may consider a two-level system where the gap of the eigenenergies (avoided crossing) may be computed directly. On the other hand, dynamically localized states only occupy a smaller area of the phase space. If the consecutive states occupy different areas of the phase space, there will be essentially no overlap between them and the couplings are exponentially suppressed. This is more likely to happen if the eigenstates are more severely localized, leading to a much weaker level repulsion.

2) I do not see how information about the level repulsion might be obtained from the SFF directly at the Heisenberg time.

3) The approach of using the mean time between collisions was used to convert the discreet transport times of the map to continuous time was applied in the previous paper cited in the manuscript. I agree with the referee that the time like variable should correspond to a continuous time but as mentioned simply using the mean time between collisions as well as taking into account the scaling of the mean level spacing with the deformation parameter gives a different scaling result for the ratio between the transport time and Heisenberg time. As mentioned in the text I believe the difference stems from the difference in the decay of correlations between the billiard map and the flow. To get a final answer one would have to compute the transport times of the diffusion of for instance angular momentum (a variable that is still well defined between collisions) in the stadium in continuous time.

4) The scaling seems to not depend on the value of g_0 even if we increase the threshold to for instance g_0=0.05

I have corrected the typos and renamed the Heisenberg time when used in the context of the units in the SFF to tau_H.

Reviewer 2 Report

The author studies the connection of the spectral form factor to dynamical localization using the stadium billiard as an example. He uses a quantum transport time extracted from the spectral form factor, using many high-lying states. The epsilon parameter of the stadium is changed over the transition from localization to ergodicity (for the considered energy range). In some cases it might indeed be an advantage to extract the transport time from purely quantum information instead from classical dynamics. This transport time is compared to the exponent of a Brody distribution fitted to the nearest neighbor level-spacing distribution, showing a linear dependence.

This is a very interesting and well written paper. It brings new insight into dynamical localization and should be published in the present form. Furthermore, it is well suitable for this special issues.

Two minor optional comments:
1) The plural 'form factors' in the title might be reconsidered.
2) There seems to be a typo at the end of line 20 ('at').

Author Response

I thank the reviewer for the careful reading of the manuscript and supporting the publication. I agree that it is better to use the singular form in the title and have corrected the typos.

Reviewer 3 Report

Spectral form factors and dynamical localization

ÄŒrt Lozej

submitted to Entropy (received 31/01/2023)

This paper investigates numerically statistical properties of a family of stadium billiards. It shows that the transport time extracted from the time-dependance of the spectral form factor can be related to the level repulsion exponent characterizing local statistical properties of the spectrum.

This is an interesting piece of work. The paper is well-written, pedagogic and well-organized. An explanation for the central result (a linear correlation between level repulsion exponent beta and transport time) is missing; however this linear correlation is rather convincing, based on careful numerical investigation, and calls for an explanation. As such, the result should be published.

I am less convinced by the functional relation between transport time and parameter epsilon: the claimed power-law decay, as presented in the inset of Fig.2, is as good as the decay observed in the main panel. Additionally, as the author stresses, the data are over less than a decade. Besides, there is no theoretical justification for the power law, nor for the exponents 1/2 and 1. Can the transport time be related to some characteristic length within the billiard?

Despite these latter limitations, I do recommend the Article for publication in Entropy.

Some minor points:
-Please perform a spell-check before submitting. There are misprints at numerous places (including in the title of Ref. [42], which is the Authors' own thesis!)
-the value of the Heisenberg time should be clarified: it takes the value 2pi (below Eq.4) or 1 (below Eq.5)
-In Fig.1 the color code does not have a legend: is it the eigenfunction value psi_i, or is it log|psi_i|, or something else?
-At the end of section 3.2, the expression "a more general relation" sounds a bit mysterious; could it be elucidated?

Author Response

I thank the reviewer for the careful reading of the manuscript and useful remarks, as well as supporting the publication.
Regarding the power law behaviour of the transport times, I agree that a more extensive numerical investigation on both the classical and quantum sides would be required to ultimately settle the question. I believe the difference from the previous power laws obtained from the momentum diffusion stem from the differences in the correlation decays in the discrete time versus the continuous time dynamics. Regardless of the power laws, the decay of the transport times is monotonic and does not compromise the central result.
I have corrected the (rather embarrassing) typos and changed the label of the Heisenberg time when measured in the SFF units to tau_H restoring consistency. I have added a comment in the caption regarding the quantity plotted, it is log(psi^2). Furthermore, I have clarified that the more general relation is referring to the possibility that the linked non-universal behaviour of the SSF and level spacings might be observed in other contexts, not only in the case of dynamical localization.

Round 2

Reviewer 1 Report

The author addressed all my remarks and clarified my questions. I recommend the article for publication in its present form.